# IP₃ receptor isoforms differently regulate ER-mitochondrial contacts and local calcium transfer

Adam Bartok[1,2], David Weaver[1], Tünde Golenár[1], Zuzana Nichtova[1], Máté Katona[1], Száva Bánsághi[1], Kamil J. Alzayady[3], V. Kaye Thomas[3], Hideaki Ando[4,5], Katsuhiko Mikoshiba[4,6], Suresh K. Joseph[1], David I. Yule[3], György Csordás[1] & György Hajnóczky [1]

Contact sites of endoplasmic reticulum (ER) and mitochondria locally convey calcium signals between the IP₃ receptors (IP3R) and the mitochondrial calcium uniporter, and are central to cell survival. It remains unclear whether IP3Rs also have a structural role in contact formation and whether the different IP3R isoforms have redundant functions. Using an IP3R-deficient cell model rescued with each of the three IP3R isoforms and an array of super-resolution and ultrastructural approaches we demonstrate that IP3Rs are required for maintaining ER-mitochondrial contacts. This role is independent of calcium fluxes. We also show that, while each isoform can support contacts, type 2 IP3R is the most effective in delivering calcium to the mitochondria. Thus, these studies reveal a non-canonical, structural role for the IP3Rs and direct attention towards the type 2 IP3R that was previously neglected in the context of ER-mitochondrial calcium signaling.

[1] MitoCare Center, Department of Pathology, Anatomy and Cell Biology, Thomas Jefferson University, Philadelphia, PA, USA. [2] Departent of Medical Biochemistry, Semmelweis University, Budapest, Hungary. [3] Department of Physiology and Pharmacology, University of Rochester, Rochester, NY, USA. [4] Lab for Developmental Neurobiology, RIKEN Brain Science Institute, Saitama, Japan. [5] Present address: Laboratory of Molecular Biomedicine for Pathogenesis, Center for Disease Biology and Integrative Medicine, Faculty of Medicine, The University of Tokyo 7-3-1 Hongo, Bunkyo-ku, Tokyo 113-0033, Japan. [6] Present address: Shanghai Institute for Advanced Immunochemical Studies, ShanghaiTech University, 201210 Shanghai, China. Correspondence and requests for materials should be addressed to G.C. (email: gyorgy.csordas@jefferson.edu) or to G.H. (email: gyorgy.hajnoczky@jefferson.edu)

In the past decade, interorganellar signaling and its patho-physiologic connections to various human diseases have received growing attention. One of the most intensely pursued paradigms has been the local communication between the endoplasmic reticulum (ER) and mitochondria at their close contacts, which are supported by direct physical links referred to as tethers[1]. ER-mitochondrial contacts are relevant for intracellular signaling through calcium, reactive oxygen species and Bcl-2 family proteins, biosynthetic pathways, membrane dynamics, and even for mitochondrial DNA replication (reviewed in ref. [2–5]). The alteration of the physical contact between the ER and mitochondria and the malfunction of the interorganellar $Ca^{2+}$ transfer was linked to various complex symptoms involved in human diseases, such as progressive cognitive and motor disorders, neurodegeneration, or cancer[6–9]. Thus, determining the structure of the interface between the ER and mitochondria, the molecules involved in the physical interaction and their regulation is expected to aid understanding of the pathogenesis of such diseases and identification of potential new drug targets. In the past decade more than 60 proteins have been localized at the contacts[5,10,11], among them the $IP_3$ receptors (IP3Rs), but their distinct roles in shaping the ER-mitochondrial contacts have never been directly addressed.

IP3Rs are large-conductance cation channels that mediate $Ca^{2+}$ release from the ER, giving rise to a cytoplasmic $[Ca^{2+}]$ ($[Ca^{2+}]_{cyto}$) signal, which controls a variety of cytoplasmic targets and is also effectively delivered to the mitochondrial matrix to regulate oxidative metabolism and cell survival[12,13]. Because the global $[Ca^{2+}]_{cyto}$ usually peaks at $\leq 1\mu M$ and the affinity of the mitochondrial $Ca^{2+}$ uptake machinery is relatively low[12], and because clamping the global $[Ca^{2+}]_{cyto}$ with a slow $Ca^{2+}$ buffer does not prevent calcium signal propagation to the mitochondria[14,15], the IP3R-mitochondrial $Ca^{2+}$ delivery has to involve a privileged local mechanism. Indeed, direct local $[Ca^{2+}]$ measurements showed the IP3R-mediated $Ca^{2+}$ release can expose the adjacent mitochondrial surface to 10–30 times larger $[Ca^{2+}]_{cyto}$ than the global $[Ca^{2+}]_{cyto}$ increase[16,17].

There are three isoforms of IP3Rs, encoded by three distinct genes, all regulated by IP3, $Ca^{2+}$, ATP, and post-translational modifications (e.g., phosphorylation and redox state), but each isoform shows different activation and deactivation patterns[18–24]. The expression pattern and subcellular distribution of the three IP3R isoforms show heterogeneity in different tissues and cell types[25–27]. Overall, the three IP3R isoforms seem to have different functions and regulation, which are expected to result in heterogeneity in their communication with mitochondria. Among the three different IP3R isoforms, IP3R1 and IP3R3 have been described to co-IP with the voltage dependent anion channels (VDAC1) in the mitochondrial outer membrane (OMM) or to facilitate $Ca^{2+}$ signal transmission from the ER to the mitochondria[28–30]. However, no systematic comparison of the three IP3R isoforms in the same cell type has been conducted.

The first vertebrate cellular model available to compare individual IP3R isoforms in the same cellular background was established in DT40 cells[31]. This model is also broadly relevant since several functional consequences of IP3R-mitochondrial communication have also been described in DT40 cells[19,32,33]. In the current study, we investigate a possible role of IP3Rs in shaping the interface between the ER and mitochondria and assess the role of each IP3R isoform in delivering $Ca^{2+}$ signals to the mitochondria in IP3R-deficient (triple KO, TKO) DT40 cells stably and evenly rescued by biochemically tagged individual mammalian IP3R isoforms. To test the broad relevance of the findings made in DT40 cells, we also employ a very recently established IP3R-deficient (TKO) HeLa cell model[34].

## Results

**Tight contacts between ER and mitochondria depend on IP3Rs.** To test whether IP3Rs are needed for the physical coupling of ER and mitochondria we first used wild-type (WT) and IP3R-deficient (TKO) DT40 cells (Supplementary Fig. 1A). In TKO cells, the $IP_3$-mediated $Ca^{2+}$ signaling was completely absent (Fig. 1a). Localization of the ER relative to the mitochondria was first visualized by organelle-targeted fluorescent proteins (ER-GFP, mito-mCherry) using a confocal microscope with an Airyscan detector to improve the optical resolution (Fig. 1b). Our attempt at colocalization analysis of ER and mitochondria at ~150–200 nm resolution could not capture any IP3R-dependent difference in the TKO vs. WT and IP3R-rescued cells. However, looking at the relative abundance of the organelles among the different horizontal (Z) planes of the cell, we found that in WT cells, this Z-distribution of ER and mitochondria showed a similar pattern, whereas in TKO cells the two patterns were less aligned (Fig. 1c). The difference was most obvious in the region close to the adherent surface, therefore we examined the ER and mitochondria co-distribution in the basal (1 μm thick) region of the cells. Values close to zero for WT cells suggest that the ER and mitochondria follow the same distribution pattern, whereas in TKO cells, the relative presence of ER is higher than that of the mitochondria close to the adherent surface (Fig. 1c, inset). This phenomenon might be caused by a stronger physical interaction between the ER and mitochondria in the presence of IP3Rs, which keep the ER co-distributed with mitochondria when the cells are polarized by an adhesion surface (Cell-Tak coated coverglass), but Airyscan imaging does not have the resolution to directly visualize <100 nm wide contacts.

To further examine the spatial relationship between ER and mitochondria in live cells, we used FRET measurements that are sensitive to distances within the 10 nm scale gap-width[16]. Outer mitochondrial membrane (OMM) and ER targeting sequences coupled with the two components of the FKBP-FRB heterodimerization system and YFP and CFP, respectively, were overexpressed in WT and TKO cells. Addition of rapamycin causes *trans* heterodimerization between interfacing FKBP and FRB domains to rapidly connect the ER-and OMM-targeted anchors. Induction of the bridge formation is initially confined to the areas where the ER and OMM were naturally close. Addition of rapamycin (100 nM) led to rapid redistribution of the majority of the CFP fluorescence to the mitochondria and an increase in the FRET between CFP and YFP (Fig. 1d). The kinetics of the formation of the bond between the linker halves was measured by the change in the ratio of the FRET and CFP signal, which was significantly faster in WT cells than in the TKO (Fig. 1d, inset). This result further indicates the dependence of more close associations between ER and mitochondria on IP3R expression.

Associations between ER and mitochondria at the ultrastructural level were analyzed in electron micrographs of WT and TKO DT40 cells (Fig. 1e). The length of ER segments within 100 nm distance from the mitochondria (OMM) was measured with high spatial resolution. Quantitative interface profiles were established by binning the interface lengths in given distance ranges between the membranes. Comparing the occurrence of interactions within given gap widths between mitochondria and ER, we found significantly higher frequency of tighter interactions in WT cells (Fig. 1f). To test whether the IP3R dependence of the ER-mitochondrial interface is not a peculiarity of the DT40 cells, we also performed ultrastructure analysis in IP3R TKO HeLa cells that have just been created and validated[34]. Similar to that in the DT40 cells, the tight interactions were more frequent in the WT than in the TKO HeLa cells (Fig. 1f). These results, together with the FRET data, provide the first direct evidence for a role of IP3Rs in the formation of the ER-mitochondrial contacts.

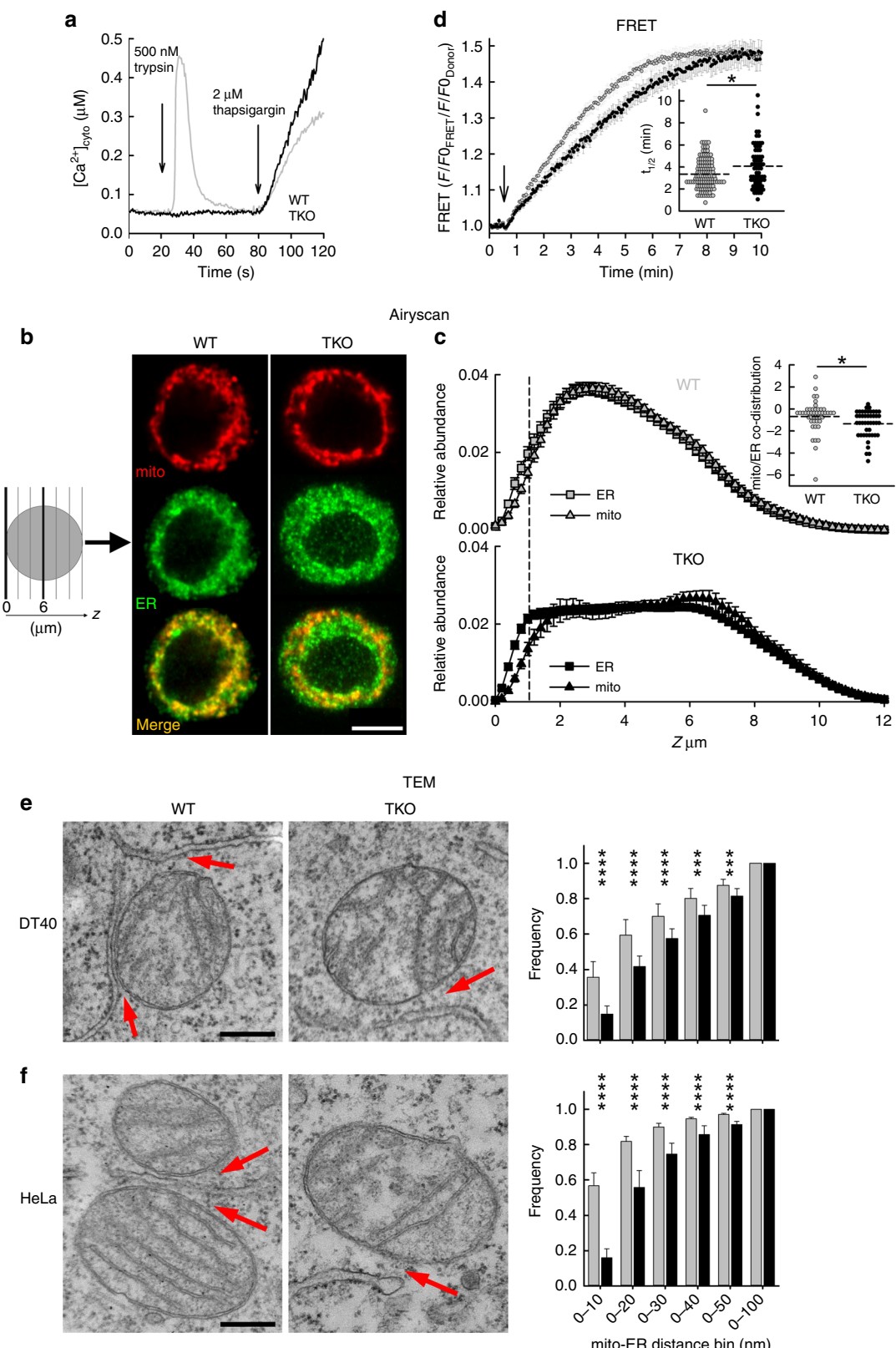

## Mammalian IP3R isoforms are functional in DT40 TKO cells.

To verify the function of each IP3R isoform, we used DT40 TKO cells rescued with an individual FLAG-tagged mammalian IP3R isoform. Comparing the rescue levels to the endogenous IP3R isoform abundance is difficult because the anti-IP3R antibodies likely recognize avian (endogenous) and mammalian (rescue)

IP3Rs with different affinities (Supplementary Fig. 1A). However, the clones selected for this study had comparable expression levels for the individual isoforms based on anti-FLAG immunoblotting (Supplementary Fig. 1B). We validated the IP$_3$ sensitivity of each clone in permeabilized cells (Supplementary Fig. 1C, D). We measured the highest IP$_3$ sensitivity for IP3R2 (EC50 = 146 nM,

**Fig. 1** Tight contacts between mitochondria and ER are more frequent in the presence of IP3Rs. **a** IP$_3$-mobilizing agonist-induced [Ca$^{2+}$]$_{cyto}$ signals in WT and TKO DT40 cells (representative traces, $N = 23$ and 35 cells, respectively, from three independent experiments). For reference, Ca$^{2+}$ mobilization through SERCA inhibition (thapsigargin) is also shown. **b** Intracellular distribution of the mitochondria (mito, mito-mCherry) and ER (ER-GFP) visualized by confocal microscopy (Z-stacks). Sections at Z = 6 μm are shown (Z = 0 at the adherent surface), see scheme. Scale = 5 μm. **c** Relative abundance (distribution of the whole-cell fluorescence amongst 60 slices taken with 0.2 μm increments) of ER and mitochondria in WT ($N = 45$) and TKO ($N = 49$) cells plotted against the Z-axis (means ± SEM from four independent experiments). **Inset** co-distribution of ER and mitochondria (Log2 converted, dashed lines: mean, *$p < 0.05$ unpaired Student's *t*-test). **d** Dynamics of the formation of tight interactions between ER and mitochondria as measured by FRET increase between CFP− (Donor, ER-targeted module) and YFP− (Acceptor, OMM-targeted module) tagged rapamycin-inducible linker pairs. FRET ratios (FRET/CFP) show different kinetics in WT and TKO cells (mean traces ± S.E.M., $N = 122$ and 71, respectively, from three independent experiments). **inset** Half-activation times (t1/2) (dashed lines: means) *$p < 0.05$, unpaired Student's *t*-test. **e, f** Example electron micrographs of mitochondria forming close contact(s) with ER (red arrows) in WT and TKO DT40 (**e**) and HeLa (**f**) cells. Scale bar = 300 nm. **Bar graphs** Quantitation of the tight associations between mitochondria and ER. In TEM images of randomly selected WT and TKO cells, for all mitochondria with ER contact (<100 nm distance) the mitochondrial perimeter (OMM) and interfacing ER membrane segments were masked and measured, binning the interface distances and returning the length of the ER subsegment that participates in an interface of a given gap-width. Bars show relative frequencies (Mean ± S.E.M of three experiments) of minimum distances providing an interface <100 nm between the ER and mitochondrion in WT (gray) and TKO (black) cells. ***$p < 0.001$, ****$p < 0.0001$ with Chi-squared test ($N = 591$ WT and 531 TKO mitochondria in DT40 and 702 WT and 632 TKO mitochondria in HeLa cells from three independent experiments). Source data are provided for panels **c**, **d**, **e**, **f** as a Source Data file

**Table 1 Functional comparison of endogenous IP3R levels to those in stable-rescued cells**

|              |                | IP3R1      | IP3R2     | IP3R3       |
|--------------|----------------|------------|-----------|-------------|
| Stable rescue| Pool size (%)  | 63 ± 1     | 82 ± 1    | 69 ± 3      |
|              | EC50 (nM)      | 213 ± 44   | 146 ± 8   | 1664 ± 492  |
| DKO          | Pool size (%)  | 71         | 74        | 34          |
|              | EC50 (nM)      | 331        | 209       | 2895        |

IP$_3$ sensitivity was measured in permeabilized cells with cuvette-based fluorimeter. IP$_3$ sensitive pool size was determined as the percentage of the maximum Ca$^{2+}$ signal measured at saturating IP$_3$ concentration relative to that at Tg addition (mean ± SEM, $N = 37$, 28, and 15 runs, respectively, from six independent experiments). EC50 values were determined by fitting sigmoidal, Hill, 3 parameter function on data points of dose-response measurements ($N = 74$, 55, and 60 points, respectively) For more details, see Supplementary Fig. 1C, D. DKO results are from[19]. Source data are provided as a Source Data file

H = 2.37). The two IP3R1 clones (R1 A and R1 B) had similar sensitivities (EC50 = 213 nM, H = 1.18 for R1 A and 218 nM, H 1.46 for R1 B). IP3R3 has the lowest sensitivity (EC50 = 1664 nM, H = 1.08) (Supplementary Fig. 1B). The IP$_3$-sensitive ER Ca$^{2+}$ pool sizes (percentage of maximal IP$_3$-induced release relative to the thapsigargin-induced Ca$^{2+}$ release) of the cell lines were in the range of 50–80%. The IP$_3$ sensitivity and pool sizes were then compared to what we recorded previously in double knockout (DKO cells) DT40 cells expressing each individual endogenous avian IP3R isoform[19] (Table 1). The EC50 values show similar patterns in both systems, though we found slightly increased sensitivities in the stable rescue system. The IP$_3$ sensitive pool size measured in the TKO rescue and DKO systems are also comparable with the exception of DKO cells expressing endogenous avian IP3R3, which had a relatively small IP$_3$ sensitive pool. In conclusion, the different mammalian IP3R isoforms in the stable rescue cells provide a suitable model for studying their function in the same cellular background. Furthermore, having the FLAG tag on each rescue allows studying their localization by the same antibody.

**Each IP3R isoform can restore tight ER-mitochondria contacts**. To test the effects of the three different IP3R isoforms on ER and mitochondrial distribution and contacts, we compared TKO cells with those stably rescued with a single, FLAG-tagged IP3R isoform. We used two clones of IP3R1 (A, B) and one clone each of IP3R2 and IP3R3. As a first approach, we examined the relative distribution of ER and mitochondria along the Z-axis of plated cells using the Airyscan detector (Fig. 2a). We found that each

IP3R isoform increased the ER-mitochondria co-distribution close to the adherent surface (0–1 μm) (Fig. 2b) as compared to that in TKO cells (Fig. 1c), similarly to what we had seen in WT cells (see Fig. 1c). The competency of each IP3R isoform to rescue the co-distribution was established by calculation of the relative sum of square values for each cell by taking the sum of the square of the difference of the relative abundances of the two organelles within each Z plane (Fig. 2c). The higher values for TKO cells indicate a greater difference in the distribution pattern of the organelles. With ultrastructural analysis, the frequencies of the occurrence of close interfaces between the ER and mitochondria in the range of 0–10 nm and 10–20 nm were higher for each IP3R isoform expressing clone compared to TKO cells (Fig. 2d), demonstrating that each IP3R isoform enhances the physical connection between the ER and mitochondria. Together these results show isoform-independent effects of IP3Rs on the relative positioning of ER and mitochondria.

**Each IP3R isoform forms clusters close to the mitochondria.** We employed three techniques to visualize directly the relationship between IP3Rs and mitochondria in stable-rescued DT40 cells. First, the localization of FLAG-tagged IP3Rs and OMM was visualized by single-molecule localization super-resolution microscopy (Fig. 3a). A cluster-based analysis was used to measure 3D distances between IP3R clusters and the surfaces of mitochondria and compared against measurements on randomly generated clusters of similar dimensions (see Methods). Minimum distances between IP3R clusters and OMM were binned in 25 nm intervals (Fig. 3b). The fractions of minimum distance values in the range of 0–25 nm, relative to all clusters within 250 nm, were significantly higher for each IP3R isoform clusters than that of random clusters, suggesting that a fraction of each IP3R isoform is targeted to the close vicinity of the mitochondria.

Next, we sought to validate these findings using 3D stimulated emission depletion (STED), super-resolution microscopy. We were able to visualize FLAG-tagged IP3R2 and IP3R3 together with the OMM marker TOM20 and to localize areas where IP3R clusters overlapped with OMM signal (Supplementary Fig. 2A, B). While the IP3R2 seemed overall to have more contacts with the mitochondria, quantitative measurements were highly variable cell-to-cell, possible due to varying IP3R expression levels. Finally, we performed immuno-EM experiments, which provide very high spatial resolution, but are limited to 2D. We found that anti-FLAG gold particle density was higher within 0–25 nm from the surface of mitochondria than in the 25–100 nm range (Supplementary Fig. 2C, D, E) in agreement with the single-

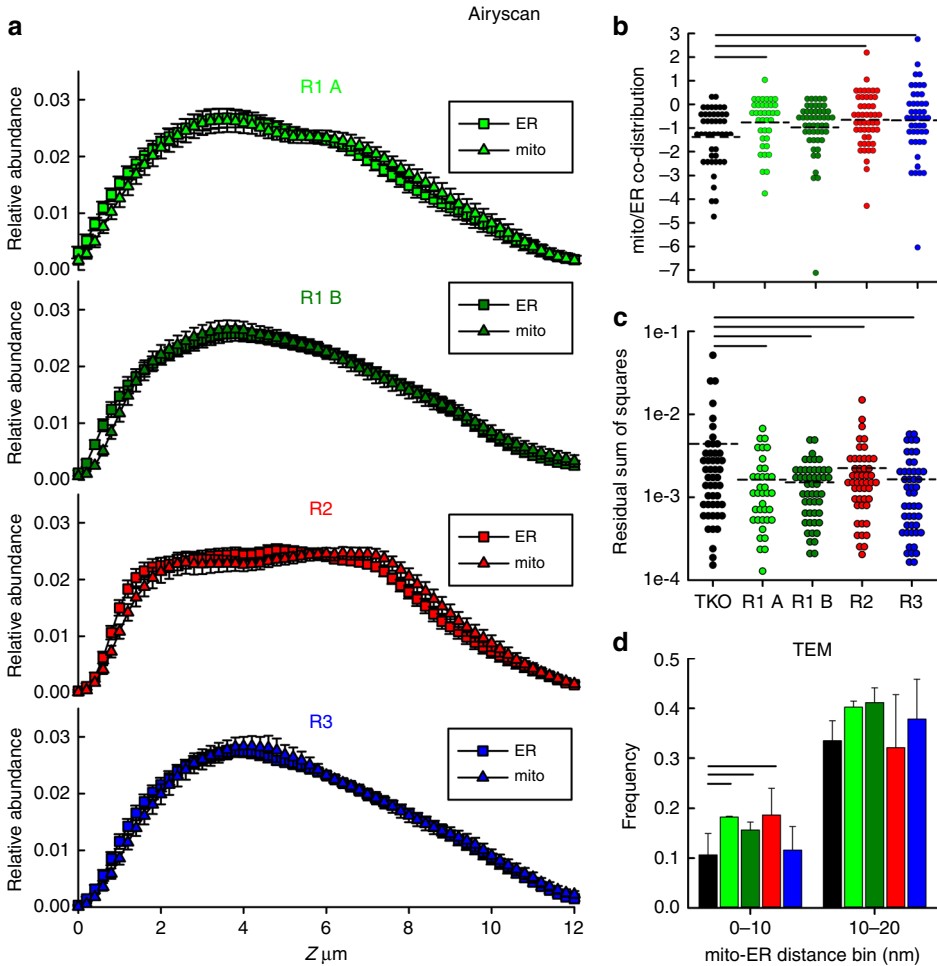

**Fig. 2** Each IP3R isoform restores ER-mitochondria co-distribution and tight interactions. **a** Relative abundance of ER and mitochondria in R1 A ($N = 38$), R1 B ($N = 50$), R2 ($N = 49$), and R3 ($N = 48$) cells plotted against the $Z$-axis (mean ± SEM from four independent experiments). **b** Co-distribution of mitochondria and ER in the 0–1 μm Z sections of each clones (Log2 converted, mean, dashed line). **c** Residual sum of squares for mitochondria and ER. Cellular averages (dashed lines) are compared. Black lines in **b**, **c** indicate $p < 0.05$ with one-way ANOVA with post-hoc Holm method. **d** TEM evaluation of the close interfaces between ER and mitochondria in the different IP3R rescue clones (same color code as in **a**). Bar graphs show relative frequencies (Mean ± S.E.M of three experiments) of minimum distances at the interfaces of <100 nm between the ER and mitochondria in each cell line. Black lines represent $p < 0.05$ with Chi-squared test ($N = 238$ TKO, 312 R1 A, 322 R1 B, 310 R2 and 375 R3 mitochondria from three experiments). Source data are provided for each panel as a Source Data file

molecule localization fluorescence imaging. Consistent with the STED results, the IP3R2 isoform seemed to be most enriched within 0–25 nm from the OMM surface (Supplementary Fig. 2E third row).

**Distributions of IP3R isoforms differ in plated cells**. To examine the intracellular localization of IP3Rs at the whole-cell level, IP3Rs were visualized using the Airyscan detector (Fig. 3c). Plotting the intracellular distribution of the IP3Rs, we found that a large fraction of the pool of IP3R1 concentrates to the adherent surface whereas IP3R2 and IP3R3 are more uniformly distributed in the cells (Fig. 3d). To quantify this difference, we calculated the relative abundance of the IP3Rs within 0–1 μm distance from the adherent surface (Fig. 3e), which showed a significant increase for IP3R1 as opposed to IP3R2 and IP3R3. The residual sum of squares for the IP3Rs and ER shows significantly higher values, thus bigger differences in the distribution patterns for both R1 A and R1 B compared to R2 and R3 rescue clones (Fig. 3f). These results suggest that a fraction of each IP3R isoform is localized to the ER-mitochondrial contacts while IP3R1s have a primary

concentration close to the adherent region of the plasma membrane.

**IP3R2>IP3R3>IP3R1 at propagating Ca$^{2+}$ signals to mitochondria**. To test the ability of the IP3R isoforms to transfer Ca$^{2+}$ from the ER to the mitochondria we measured cytoplasmic and mitochondrial matrix [Ca$^{2+}$] ([Ca$^{2+}$]$_{cyto}$ and [Ca$^{2+}$]$_{mito}$) levels, simultaneously in intact cells. To trigger IP$_3$-dependent ER Ca$^{2+}$ release we stimulated the cells with trypsin, which activates the IP$_3$-linked protease-activated receptors (PAR). Addition of trypsin was followed by cytoplasmic and closely coupled mitochondrial Ca$^{2+}$ signals in each IP3R isoform expressing cell line (Fig. 4a). Baseline [Ca$^{2+}$]$_{cyto}$ levels were not different in the different IP3R rescue clones (Fig. 4b left). Cytoplasmic and mitochondrial Ca$^{2+}$ signals for R1 A and R1 B clones showed slower rise and decay kinetics and lower peak values compared to those in the R2 and R3 clones (Fig. 4a–c). To measure and compare the time course of the ER-mitochondrial Ca$^{2+}$ transfer for the different IP3R isoforms we determined the coupling time as the difference of the half-activation time of the [Ca$^{2+}$]$_{mito}$ and

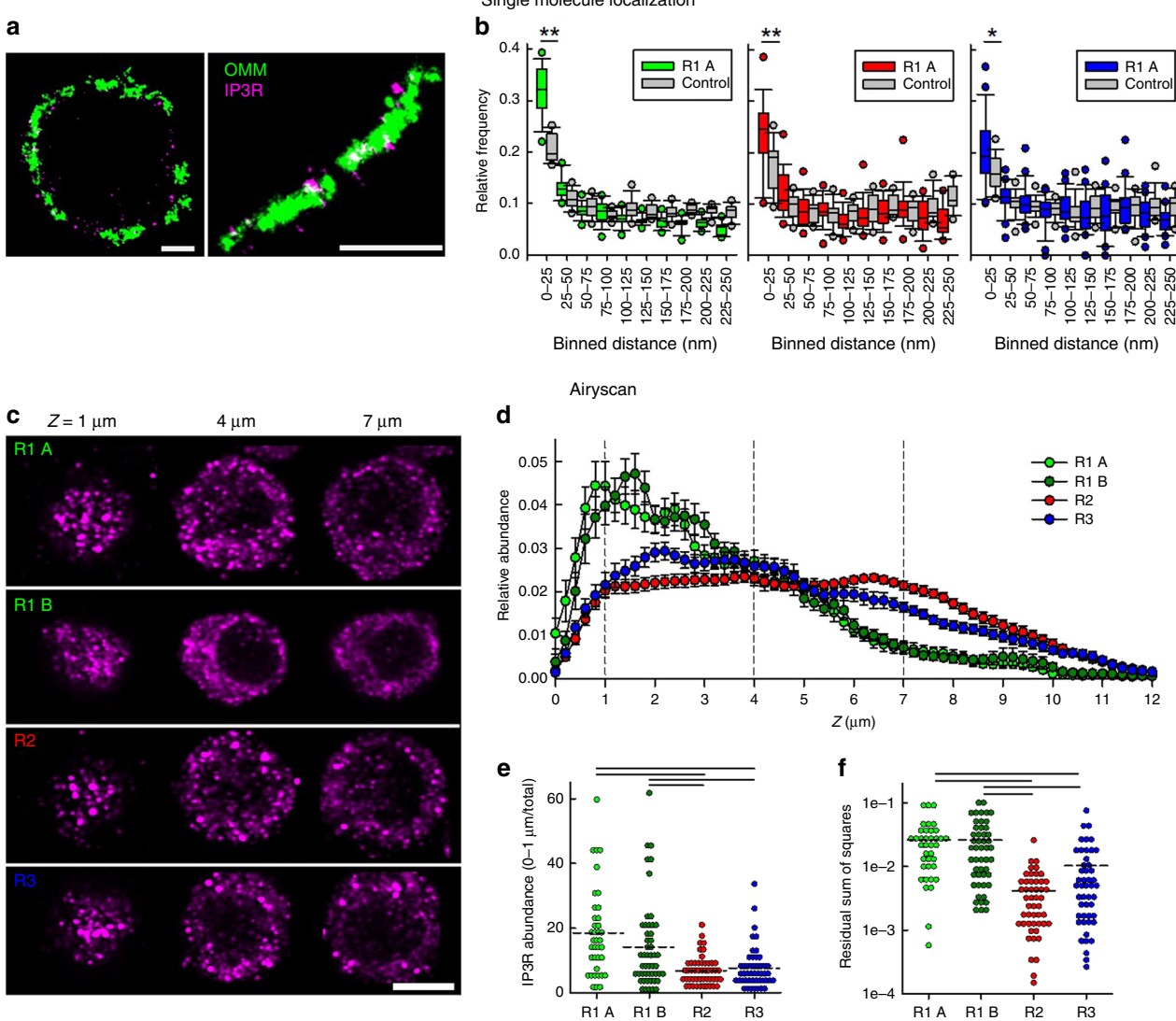

**Fig. 3** Each IP3R isoform forms clusters near mitochondria, but their subcellular distributions vary. **a** OMM (Tom20) and IP3Rs visualized by Vutara 352 STORM based super-resolution (single-molecule imaging) system. 2D projection of Z = 200 nm thickness is shown. Bars = 2 μm. **b** Shortest distances between IP3R and OMM clusters were measured in 3D. Relative frequencies of minimum distances are shown within 0–250 nM (median line, box: 25–75%, error bars 10–90% and individual outliers as dots N = 15, 18, 24 cells, respectively, from three independent experiments). For reference, the same measurements were performed using mock randomly generated clusters in place of the IP3R clusters (control) (N = 14, 13, 19, respectively). *p < 0.05, **p < 0.01, unpaired Student's t-test. **c** Anti-FLAG immunofluorescence in IP3R-FLAG expressing cells in Z sections at 1, 4, and 7 μm. Scale bar = 5 μm. **d** Relative abundance of IP3Rs in R1 A (N = 38), R1 B (N = 50), R2 (N = 49), and R3 (N = 48) cells plotted against the Z-axis. **e** Cumulative abundance of IP3Rs in the 0–1 μm Z sections of each cells (mean, dashed lines from four independent experiments). Black lines indicate p < 0.05 with one-way ANOVA with post-hoc Dunn's method. **f** Residual sum of squares for mitochondria and IP3Rs (Mt-IP3R). Cellular averages (dashed lines) are compared. Black lines represent p < 0.05 with one-way ANOVA with post-hoc Dunn's method. Source data are provided for panels **B**, **d**, **e**, **f** as a Source Data file

$[Ca^{2+}]_{cyto}$ signals measured in individual cells. We measured the fastest coupling in the R2 followed by R3 then R1 clones (Fig. 4d).

To test whether fast ER-mitochondrial $Ca^{2+}$ coupling also applies to the endogenous IP3R2 of the DT40 cells, we compared $[Ca^{2+}]_{cyto}$ and $[Ca^{2+}]_{mito}$ responses in permeabilized WT and the endogenous R2 expressing double (R1, R3) knockout (cR2 DKO) cells, and found higher maximal $[Ca^{2+}]_{mito}$ levels in cR2 DKO cells following direct stimulation by IP3 (Supplementary Fig. 3A). We also compared the efficiency of the ER-mitochondrial $Ca^{2+}$ transfer by comparing the percentage of cells with rapid $[Ca^{2+}]_{mito}$ rises among WT, cR1, cR2, and cR3 clones, and cR2 cells also stood out in this comparison (Supplementary Fig. 3B).

Thus, both the endogenous avian IP3R2 and the mammalian IP3R2 used for rescue are highly efficient in ER-mitochondrial $Ca^{2+}$ transfer in DT40 cells.

To further investigate the role of the different IP3R isoforms in the $Ca^{2+}$ transfer from ER to mitochondria we used photo-uncaging of IP3 (Fig. 5a). Because IP3R1 were localized at least in part, close to the plasma membrane we suspected that IP3R1 might recruit more effectively $Ca^{2+}$ entry than the other IP3R isoforms. Thus, to assess the respective contributions of ER $Ca^{2+}$ release and $Ca^{2+}$ entry, IP3 uncaging was performed both in the presence and in the absence of $Ca^{2+}$ in the extracellular medium. Because the wavelength of the (UV) uncaging light overlaps with

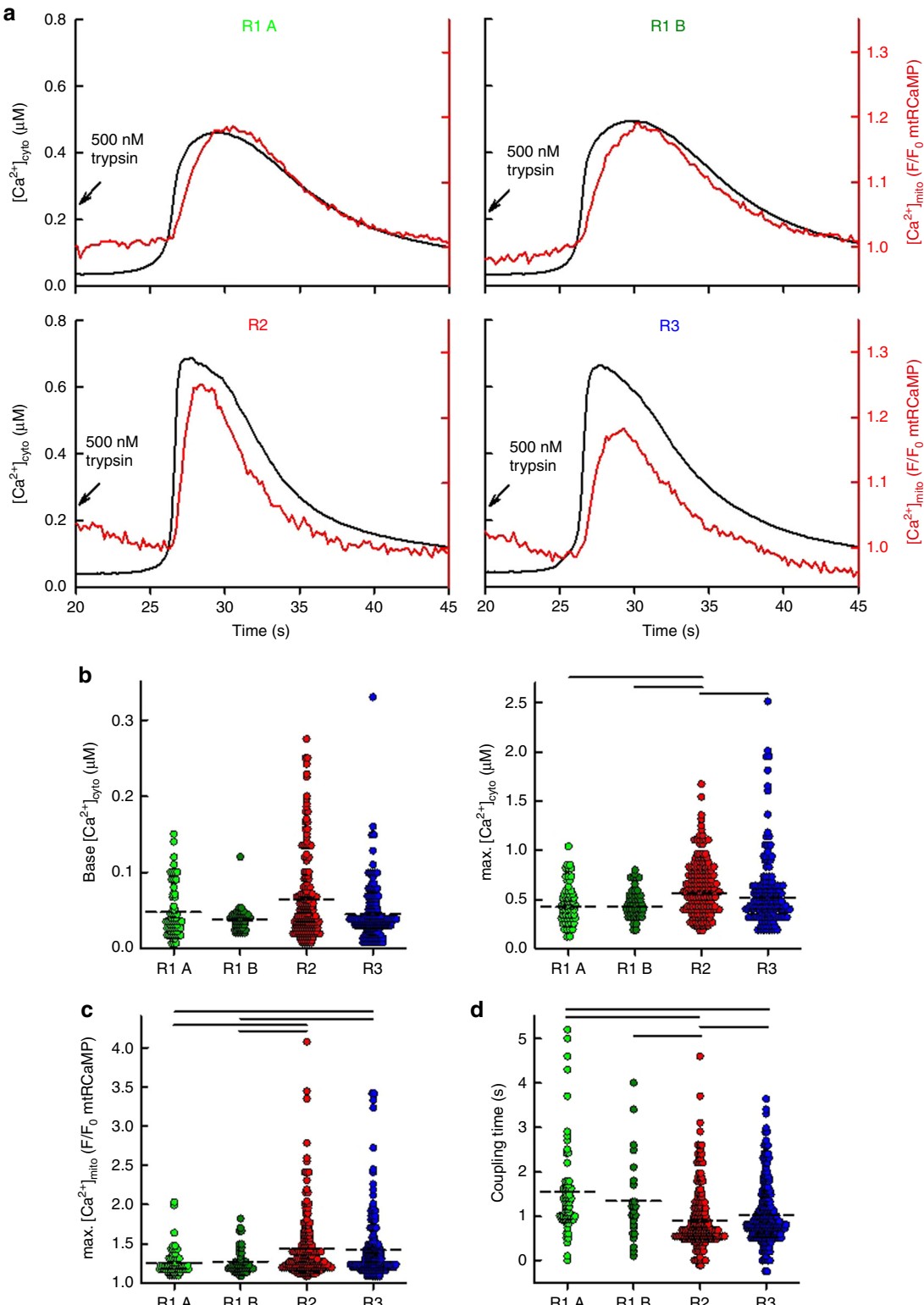

**Fig. 4** IP3R2 is the most efficient coupler of agonist-induced Ca$^{2+}$ signal. **a** To monitor Ca$^{2+}$ transfer from ER to mitochondria [Ca$^{2+}$]$_{mito}$ and [Ca$^{2+}$]$_{cyto}$ were imaged simultaneously using a wide-field EMCCD imaging setup in intact cells transfected with mitochondrial matrix-targeted mtRCaMP and loaded with fura2. IP$_3$ dependent Ca$^{2+}$-release was stimulated with 500nM trypsin. Average traces of [Ca$^{2+}$]$_{cyto}$ (black) and [Ca$^{2+}$]$_{mito}$ (red) are shown. Individual traces ($N$ = 24, 30, 19, and 21 cells, respectively, in five independent experiments) were synchronized to the half-activation time of the [Ca$^{2+}$]$_{cyto}$ signal. **b** Baseline cytoplasmic Ca$^{2+}$ levels (left panel) and peak values (right panel) followed by trypsin stimulation (mean, dashed lines). **c** Maximal [Ca$^{2+}$]$_{mito}$ responses (mean, dashed lines). **d** Coupling times (mean, dashed lines): difference of the half-activation times of the [Ca$^{2+}$]$_{cyto}$ and [Ca$^{2+}$]$_{mito}$ signals for individual cells. **b–d** Black lines indicate $p < 0.05$ with one-way ANOVA with post-hoc Dunn's Method, $N$ = 61, 49, 179, and 152 cells, respectively, from four independent experiments. Source data are provided for panels **b**, **c**, **d** as a Source Data file

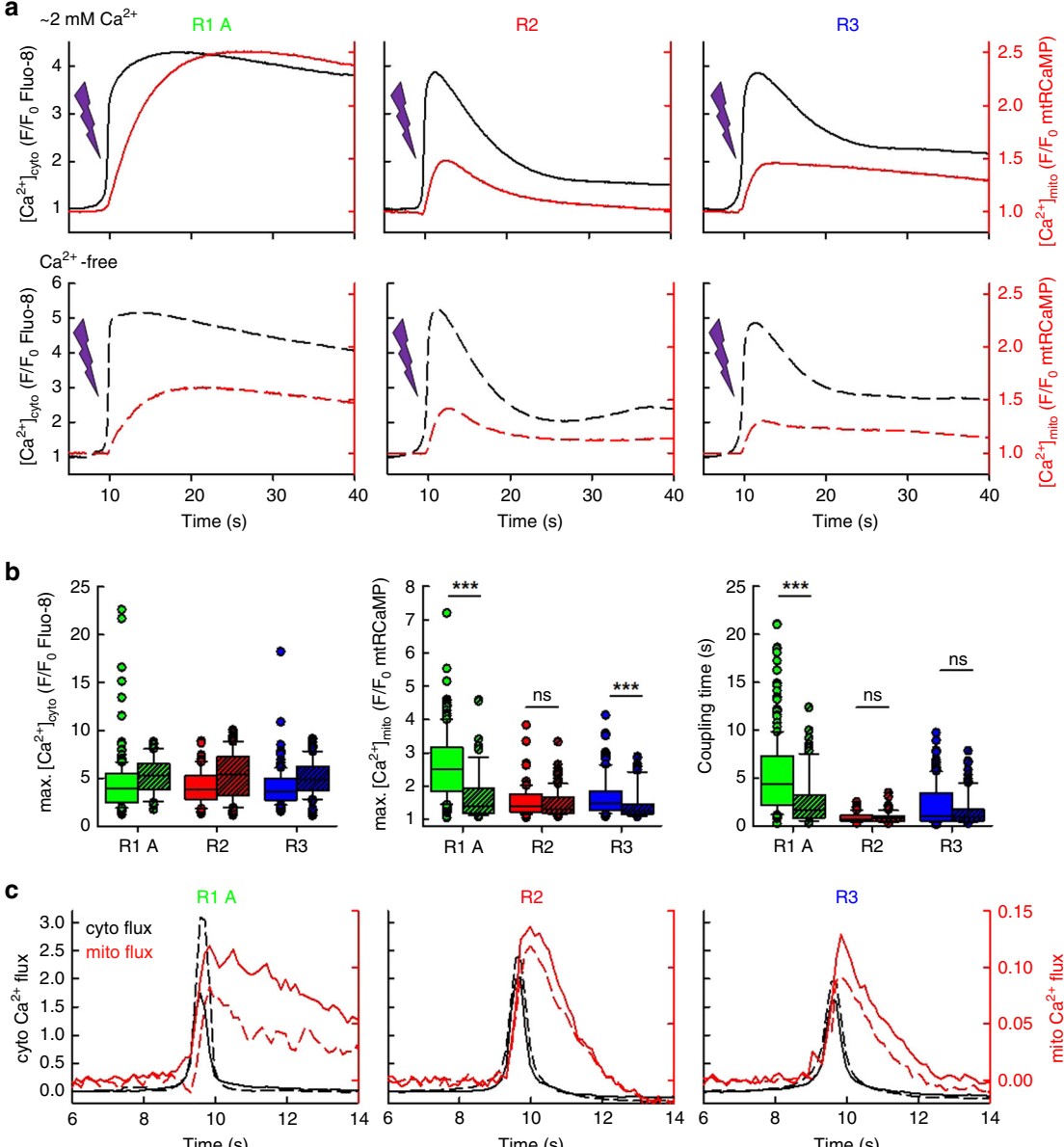

**Fig. 5** IP3R2 couples mitochondria best to IP$_3$-induced Ca$^{2+}$ signal. **a** DT40 cells were transfected with mtRCaMP then loaded with caged IP$_3$ and fluo8/AM. Calcium signals were evoked by photo-uncaging of IP$_3$ (lightning bolts). [Ca$^{2+}$]$_{cyto}$ and mitochondrial [Ca$^{2+}$]$_{mito}$ signals were measured simultaneously in individual cells. Synchronized, mean Ca$^{2+}$ signals in the presence (top row, $N = 215$, 91, and 131 cells, for R1A, R2, R3, respectively, from three independent experiments) and absence (bottom row, $N = 77$, 95, and 83 cells, respectively, for R1A, R2, R3) of Ca$^{2+}$ in the extracellular medium. **b** Box plots show quantification of maximal [Ca$^{2+}$]$_{cyto}$, maximal [Ca$^{2+}$]$_{mito}$ signals and coupling times compared in the presence and absence (dashed bars) of Ca$^{2+}$ in the extracellular medium (median line, box: 25–75%, error bars 10–90% and individual outliers as dots). ***$p < 0.001$ with Mann–Whitney Rank Sum Test. **c** Mean Ca$^{2+}$ fluxes calculated as $F_t$-$F_{t-0.5s}$ for $F_{fluo8}$ and $F_{mtRCaMP}$ in the presence (solid lines) and absence (dashed lines) of Ca$^{2+}$ in the extracellular medium. Source data are provided for panel **b** as a Source Data file

the excitation wavelengths of fura2, [Ca$^{2+}$]$_{cyto}$ was measured with a single wavelength probe, Fluo-8. When Ca$^{2+}$ was present, the peak amplitudes of [Ca$^{2+}$]$_{cyto}$ signals were similar for each IP3R isoforms (Fig. 5b left); however, compared to R2 and R3, the R1 clone displayed a sustained, slowly decaying [Ca$^{2+}$]$_{cyto}$ rise, which showed faster kinetics in the absence of Ca$^{2+}$. We measured similar peak [Ca$^{2+}$]$_{cyto}$ values for each isoform in the absence of extracellular Ca$^{2+}$. Peak values appeared universally higher in the absence of Ca$^{2+}$ when the response was normalized to the baseline (F/F0). To understand this phenomenon, we measured baseline [Ca$^{2+}$]$_{cyto}$ levels in R1 cells with fura2, which, as opposed

to Fluo-8, allows the precise calibration of the measured Ca$^{2+}$ concentration (Supplementary Fig. 4). We measured lower baseline values in the absence of Ca$^{2+}$, which explains why the baseline-normalized peaks were relatively high in this condition. In the presence of Ca$^{2+}$, the [Ca$^{2+}$]$_{mito}$ signal amplitudes were significantly higher for R1, compared to R2 and R3, with a sustained elevated phase, whereas in the absence of Ca$^{2+}$ its peak amplitude was markedly decreased. (Fig. 5b middle). Coupling times measured in the presence of Ca$^{2+}$ reflect a pattern similar to what we measured with agonist-induced stimulation (see Fig. 4), fastest coupling for R2 and the slowest for R1, however, in

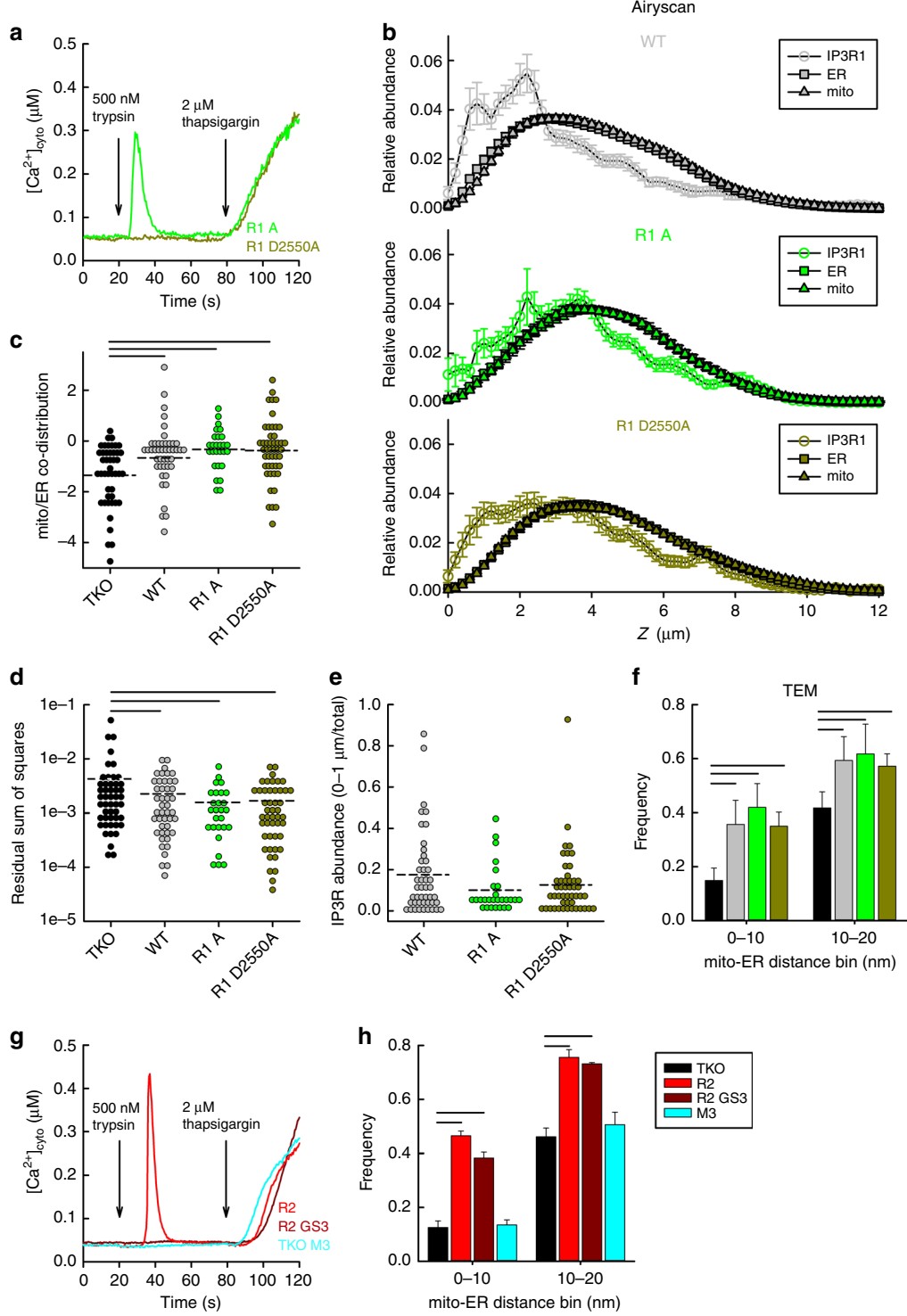

the absence of extracellular $Ca^{2+}$, R1 coupling gets faster (Fig. 5b right). We then calculated the first derivatives of the $[Ca^{2+}]_{cyto}$ and $[Ca^{2+}]_{mito}$ time courses to compare the cytoplasmic and mitochondrial $Ca^{2+}$ fluxes (Fig. 5c) and found a marked decrease in the mitochondrial $Ca^{2+}$ flux in the absence of extracellular $Ca^{2+}$ for R1 (Fig. 5c left). Such changes were minor for IP3R2 and IP3R3.

Store-operated $Ca^{2+}$ entry (SOCE) has been used to globally expose mitochondria to elevated $[Ca^{2+}]_{cyto}$ in intact cells. We found that in store-depleted cells, the presence of any IP3Rs facilitates SOCE, especially IP3R1 (Supplementary Fig. 5A-C).

However, the SOCE (non-IP3R) associated $[Ca^{2+}]_{mito}$ rise shows no sign of IP3R-dependent sensitization (Supplementary Fig. 5D).

In summary, the simultaneous $[Ca^{2+}]_{cyto}$ and $[Ca^{2+}]_{mito}$ measurements suggest that IP3R2 and IP3R3 effectively engage rapid ER-mitochondrial $Ca^{2+}$ transfer, whereas IP3R1 can support sustained $Ca^{2+}$ delivery to the mitochondria mostly from enhanced $Ca^{2+}$ entry in DT40 cells.

**Tight contacts do not depend on $Ca^{2+}$ flux through IP3Rs.** We next set out to determine if the localization and the structural behavior of IP3Rs was dependent on the $Ca^{2+}$ transfer between

**Fig. 6** Ca$^{2+}$ flux through the IP3R is dispensable for supporting the ER-mitochondrial contacts. **a** IP$_3$ dependent Ca$^{2+}$ signaling was tested as in Fig. 1a. in TKO cells rescued with WT rIP3R1 (R1 A) and with a pore-dead mutant IP3R1 (R1 D2550A). Representative cell traces are shown ($N = 30$ and 30 cells, respectively, from three independent experiments). **b** Relative abundance (Mean ± SEM from four independent experiments) of ER, mitochondria and IP3R1 in WT ($N = 45$), R1 A ($N = 27$) and R1 D2550A ($N = 47$) cells plotted against the Z-axis. **c** Co-distribution (mean, dashed lines) of ER and mitochondria calculated as in Fig. 1c. **d** Residual sum of squares for mitochondria and ER (mean, dashed lines). **e** Cumulative abundance of IP3Rs in the 0–1 μm Z sections of each cells (mean, dashed lines). Black lines indicate $p < 0.05$ with one-way ANOVA with post-hoc Holm's Method. **f** Bar charts show the relative frequencies (Mean ± S.E.M of three experiments) of the occurrence of <100 nm interface gap widths between the ER and mitochondrion ($N = 531$, 591, 477, and 561 mitochondria for TKO, WT, R1 A, and R1 D2550A, respectively). Black lines represent $p < 0.05$ with Chi-squared test. **g** IP$_3$ dependent Ca$^{2+}$ signaling in TKO cells rescued with WT IP3R2, a pore-dead IP3R2 (R2 GS3) and muscarinic acetylcholine receptor M$_3$ (M3). Representative traces are shown ($N = 164$, 189, and 197 cells, respectively, from three independent experiments). **h** Relative frequencies (Mean ± S.E.M of three experiments) of the occurrence of interface gap widths providing an interface <100 nm between the ER and mitochondrion ($N = 394$ (TKO), 268 (R2), 522 (R2 GS3), and 390 (M3) mitochondria were analyzed from three independent experiments). Black lines represent $p < 0.01$ with Chi-squared test. Source data are provided for panels **b**, **c**, **d**, **e**, **f**, **h** as a Source Data file

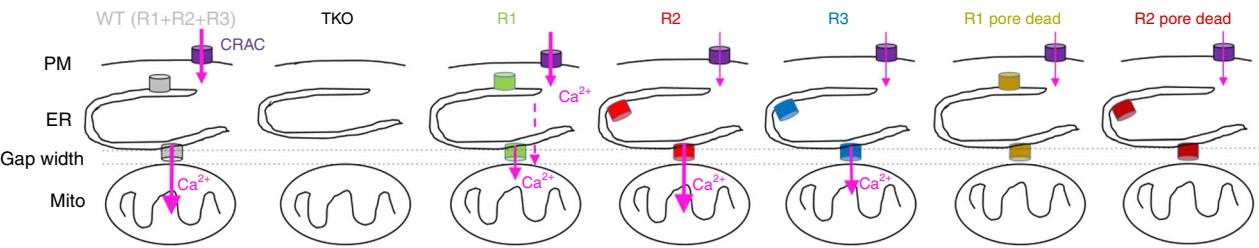

**Fig. 7** Scheme of the structural and functional properties of the individual IP3R isoforms in supporting the ER-mitochondrial interface and mitochondrial Ca$^{2+}$ uptake

ER and mitochondria. We compared WT and TKO to cells stably expressing a functional (R1 A) and a non-conductive pore mutant clone of IP3R1 (R1 D2550A, pore-dead) that fails to mediate Ca$^{2+}$ release[35] (Fig. 6a). The co-distribution of ER and mitochondria in the 0–1 μm region of each IP3R expressing cell lines was significantly higher than that measured in the TKO cells (see Fig. 1 and 2) (Fig. 6b, c). Using an anti-IP3R1 antibody[36] to examine the intracellular distribution of the receptors, the endogenous IP3R1 in WT cells, the mammalian R1 and the pore-dead R1 expressed in TKO cells were similarly concentrated towards the adherent surface (Fig. 6b, d, e). Ultrastructural measurements using TEM showed significantly higher frequency of the occurrence of 0–10 and 10–20 nm gap widths in all WT, R1 A and R1 D2550A, compared to TKO cells (Fig. 6f). To test whether our hypothesis can be extended to IP3R2s, we used a functional R2, a pore-dead R2 (R2 GS3) and a muscarinic acetylcholine receptor M$_3$ (M3) expressing TKO clone, In both R2 GS3 and M3 clones IP$_3$-dependent Ca$^{2+}$ signaling was absent (Fig. 6g). In the ultrastructure study we found that the occurrence of 0–10 and 10–20 nm gap widths were similarly higher in R2 and also R2 GS3 cells than that in the TKO cells, whereas M3 cells, negative control, were not different from the TKO. These results indicate that neither in forming tight interactions between ER and mitochondria nor in regulating the organelles' subcellular distributions do the IP3Rs' roles depend on Ca$^{2+}$ flux through the receptors.

## Discussion

We have examined the properties of the three mammalian IP3Rs in the context of the structural organization and Ca$^{2+}$ signal transmission between the ER and mitochondria. This is important because no previous study has tested the dependence of the ER-mitochondrial contact structure on the presence of any IP3Rs and no systematic assessment of the individual isoforms has been performed in terms of localization to the contacts or supporting

local Ca$^{2+}$ signaling. Separately, the different isoforms all have been implicated in mitochondria-ER communication in various tissues/cell types. IP3R1 was the first shown to be relevant to ER-mitochondrial Ca$^{2+}$ coupling by forming a complex with VDAC1 and Grp75[28]. Both IP3R1[37,38] and IP3R2[39] have been localized to the mitochondria-associated membranes (MAM) in cardiomyocytes. Studies have reported that IP3R3 (1) is a marker of the ER-mitochondrial contacts, (2) is required for apoptotic mitochondrial Ca$^{2+}$ signals[29,30], (3) concentrates to the MAMs where it is stabilized by sigma-1 receptors[40] or BAP1[6] and (4) similarly to IP3R1, co-IPs with VDAC1[29,41]. However, in most cases IP3R3 has been put forward in contrast with IP3R1 but not IP3R2, or without validating the distribution/contribution of the other isoforms[40,41].

The model system we chose was the chicken B cell (DT40) line that endogenously expresses all three IP3R isoforms and is available without any IP3Rs (TKO). This model has been widely used to evaluate the IP3R-dependence of various cellular functions[31,32,42–45]. Furthermore, DT40 cells expressing only one IP3R isoform are available (with deletion of any combinations of two IP3Rs, DKO), as well as TKO cells rescued with each mammalian IP3R isoforms, providing the possibility of side by side comparison of the individual IP3Rs in the same cellular environment[19,46].

We demonstrated that the presence of IP3Rs affects the physical associations between the ER and mitochondria using techniques that go beyond the resolution limit of conventional optical microscopy. In live cells, FRET measurement of the tight ER-mitochondrial associations showed dependence on IP3Rs (Fig. 1d). In fixed cells, electron microscopy revealed IP3R-dependent differences in the ER-mitochondrial interfaces primarily of 0–20 nm gap-width range (Fig. 1e, f). This finding was also extended to HeLa cells (Fig. 1e, f). The shortage of tight contacts in the TKO cells was reversed by stable rescue by any single mammalian IP3R isoform (Fig. 2). Thus, IP3Rs assist in maintaining close contacts either as a tethering species or as a

promoter of tether formation by other proteins. Further, enhanced resolution confocal microscopy (Fig 1b, c) revealed IP3R-dependent variances in the relative Z distributions of the ER and mitochondria, which we hypothesize may be related to the physical associations. Plating the cells on the glass coverslips creates an active adherent surface and the polarization of the cells is followed by the rearrangement of the organelles. During this process, owing to the lack of the IP3R-dependent structural support between the ER and mitochondria, a higher fraction of the ER would be available to concentrate close to the adherent surface in the TKO cells as compared to WT cells.

Recent reports suggest that in addition to VDAC1[28], other ER-mitochondrial linker species might have supporting role in the interaction of IP3Rs with the mitochondrial outer membrane (PTPIP51 for IP3R3;[41] FUNDC1 for IP3R2 in the cardiac muscle[39]. Furthermore, the IP3R-dependent gap-width measurement provided structural evidence that IP3Rs are sufficiently close to the OMM that the IP3R-mediated $Ca^{2+}$ release can create a high $[Ca^{2+}]$ nanodomain on the mitochondrial surface[16,17]. In addition, it revealed that each mammalian isoform can replace the endogenous avian IP3Rs in DT40 cells. One intriguing extension of this finding is that a pore-dead IP3R1 and IP3R2 could also rescue the close ER-mitochondrial contacts and the z-axis co-distribution of the organelles (Fig. 6), indicating that $Ca^{2+}$ release is not necessary to supporting these IP3R-dependent structural reorganizations. We speculate that a scaffold[5] or tether forming function[28] of the IP3R is central for keeping mitochondria close to the ER. The competency of each IP3R isoform, including the $Ca^{2+}$ release incompetent mutant to support close ER-mitochondrial contacts is illustrated in the scheme in Fig. 7.

Previous studies suggested that the ER-mitochondrial $Ca^{2+}$ transfer is mediated by IP3R1 or IP3R3 isoforms[28–30]. However, these studies did not consider IP3R2 and did not address the important question of how effectively each IP3R isoform would support the local communication in the same cellular paradigm. We tested simultaneously $[Ca^{2+}]_{cyto}$ and $[Ca^{2+}]_{mito}$ signals for each mammalian IP3R isoform in isolation in the same cell type during both agonist-treatment and direct $IP_3$ mobilization (Figs. 4, 5). In our system, each IP3R isoform was able to mediate mitochondrial $Ca^{2+}$ signaling, in an isoform specific manner (see purple arrows in Fig. 7). Strikingly, the IP3R-mediated $Ca^{2+}$ release was most closely followed by a $[Ca^{2+}]_{mito}$ rise for IP3R2, followed by IP3R3 and then IP3R1 (Figs. 4 and 5). Notably, the DT40 clones selected for this comparison expressed the different IP3R isoforms at comparable levels and so the above order didn't result from differences in the expression levels. Furthermore, the $Ca^{2+}$ release flux analysis showed similar peaks for IP3R2 and IP3R3 (Fig. 5). Lastly, the frequency of tight ER-mitochondrial contacts was also similar for the different rescue clones (Fig. 2). Thus, the most likely explanation for the superiority of the IP3R2 would be that on the nm scale, IP3R2s are enriched closest to the mitochondria. Nevertheless, IP3R2 has to be given more attention in the context of ER to mitochondria $Ca^{2+}$ delivery.

Among the three IP3R isoforms, IP3R1 showed a unique mediation of the mitochondrial $Ca^{2+}$ uptake, which was predominately dependent on $Ca^{2+}$ entry. Since mitochondrial $Ca^{2+}$ uptake activation by the SOCE itself did not seem to be affected by any of the IP3R isoforms (Supplementary Fig. 5D), it is likely that the dependence of IP3R1-linked $Ca^{2+}$ delivery to the mitochondria on SOCE was indirect, e.g., due to more effective SOCE-mediated refilling of ER segments at the adherent surface where IP3R1 but not IP3R2/3 seemed to concentrate (see scheme in Fig. 7). Indeed, the role of plasma membrane-ER junction localized IP3R1 in mediating SOCE was reported[47]. Our results show that IP3R1 maintains sustained cytoplasmic calcium signal, which feeds more $Ca^{2+}$ to the mitochondria as compared to IP3R2 and

IP3R3. Complementing this result, in store-depleted cells every IP3R isoform facilitated the $Ca^{2+}$ influx through the plasma membrane but IP3R1 was more effective in this regard than IP3R2 or IP3R3 (Supplementary Fig. 5). Thus, IP3R1 is less effective to direct $Ca^{2+}$ released from the ER to the mitochondria than IP3R2 or IP3R3 as claimed recently[29,30] but it is the IP3R isoform that most effectively promotes SOCE, which provides a sustained contribution to mitochondrial $Ca^{2+}$ uptake. Thus, in contrast to previous reports, IP3R2 couples mitochondria best to pure ER $Ca^{2+}$ release, but when ER store $Ca^{2+}$ depletion engages $Ca^{2+}$ entry, IP3R1s are most effective in promoting SOCE and thereby sustained mitochondrial $Ca^{2+}$ uptake.

## Methods

Reagents and chemicals were from Fisher Scientific or Sigma–Aldrich unless specified otherwise.

**Cells.** Wild-type (WT) and IP3R-deficient (TKO, cR1 DKO, cR2 DKO, and cR3 DKO) DT40 cells were provided by Prof. Tomohiro Kurosaki (RIKEN Research Center for Allergy and Immunology, Yokohama, Japan) and were cultured in suspension, in RPMI 1640 medium with glutamine (Gibco, Cat# 11875093) supplemented with penicillin/streptomycin, 2 mM L-glutamine, 10% FBS (all from GIBCO, Life Technology), 1% chicken serum (Gibco, Cat# 16110082) and 0.4% 2-mercaptoethanol in 5% CO2 and 95% air at 40 °C. DT40 cells stably expressing any IP3R or M3 construct were produced by David I. Yule, (University of Rochester, Rochester, NY) and were kept under the same conditions as above but in medium supplemented with 2 mg/ml G418 (Enzo Cat# ALX-380-013-G005).

Wild-type (WT) and IP3R-deficient (TKO) HeLa cells were provided by Katsuhiko Mikoshiba (RIKEN Brain Science Institute, Saitama, Japan). Cells were grown in Dulbecco's modified Eagle's medium (DMEM), with 4.5 g/l glucose (Gibco, Cat# 11965118) supplemented with 10% FBS, 2 mM glutamate, 100 units/ml penicillin/streptomycin (all from GIBCO, Life Technology),

**Transfection.** DT40 cells ($7.5 \times 10^6$ cells in 350 μl medium) were electroporated (BTX ECM 830) with the following constructs: mitoRCaMP, mito-mCherry, ER-GFP, ER linker, OMM linker (15 μg cDNA each) using cuvettes with 4 mm gap-width with, 230 V in two, 25 ms pulses with 3 s interval then rapidly diluted in 10 ml medium and kept at 40 °C for 24 h.

**Western blot.** All reagents used for SDS-PAGE were obtained from Bio-Rad. Harvested cells were washed once with ice-cold PBS then solubilized in cell lysis buffer containing 10 mM Tris-HCl, 10 mM NaCl, 1 mM EGTA, 1 mM EDTA, 1 mM NaF, 20 mM $Na_4P_2O_7$, 2 mM $Na_3VO_4$, 1% Triton X-100 (v/v), 0.5% sodium deoxycholate (w/v), and 10% glycerol with a mixture of protease inhibitors (Roche). After 30 min of incubation on ice lysates were cleared by centrifugation at $16,000 \times g$ for 10 min at 4 °C then transferred into fresh tubes. Protein concentrations were quantified with DC protein assay kit (Bio-Rad). Proteins were resolved on 5–7.5% SDS-PAGE gels then transferred to nitrocellulose membranes (0.45 μm, Bio-Rad). Rabbit polyclonal antibodies recognizing IP3R1 (CT1) (1:5000) raised against the C-terminal 19 amino acid (aa) residues of rat IP3R1 and IP3R2 (NT2) (1:200) raised against N-terminal 320–338 aa residues in mouse IP3R2 (custom made, Pocono Rabbit Farms and Laboratories), mouse monoclonal antibody against 22–230 aa residues of human IP3R3 (1:1000, BD Transduction laboratories Cat# 610312), Calnexin rabbit polyclonal antibody (1:1000, Enzo Cat# ADI-SPA-860), and mouse monoclonal Flag antibody (1:5000, Sigma Cat# F1804-200UG) were used for immunoblotting. Corresponding mouse and rabbit HRP-linked secondary antibodies were obtained from Cell Signaling (1:5000, Anti-mouse IgG, HRP-linked Antibody Cat#7076, Anti-rabbit IgG, HRP-linked Antibody Cat#7074). Proteins were detected with chemiluminescence using HRP substrate for CCD imaging (Azure Biosystems).

**Transmission electron microscopy.** DT40 cells were fixed and embedded using the following protocol: Cells were fixed using 2% glutaraldehyde, then with 1% OsO4 and stained with 0.5% uranyl acetate, pelleted in 2% agarose (Sigma–Aldrich, Type IX ultra low geling temperature), dehydrated in a dilution series of acetone/water, and embedded in Spurr's resin (Electron Microscopy Sciences). The sections were examined with a JEOL JEM1010 TEM fitted with a side-mounted AMT XR-50 5Mpx CCD camera or an FEI Tecnai 12 TEM fitted with a bottom-mounted AMT XR-111 10.5 Mpx CCD camera. Whole-cell images were taken with ×6700–14,000 (JEOL) or ×4400–6500 (FEI) direct magnification. Mitochondria were imaged using ×40,000 (JEOL, 1.5 nm/px resolution) or ×15,000 (FEI, 1.3 nm/px resolution) magnification[48]. Interfaces between ER and mitochondria were analyzed using a custom Image J plugin/macro written by DW.

For immuno-EM experiments the following protocol was used: Plated DT40 cells were fixed with 5% paraformaldehyde followed by quenching, permeabilization, blocking, and labeling with primary antibody (dilution: 1:1000, Sigma Cat# F1804-

200UG), followed by secondary blocking and incubating with secondary antibody anti-mouse IgG Nanogold (dilution: 1:50 Nanoprobes, Cat#2001–0.5ml). Silver enhancement (HQ Silver, Nanoprobes, Cat#2012–45 mL) was applied, followed by further fixation and contrasting with uranylacetate. Samples were dehydrated in grade of alcohol and embedded in Durcupan (Electron Microscopy Sciences). After polymerization, ultrathin sections (65–80 nm) were prepared using by ultramicrotome (Leica UTC, Diatome knife, Diatome) and studied with FEI Tecnai 12 electron microscope. Images were taken at ×3200–15,000 magnification.

**Ca²⁺ imaging experiments**. DT40 cells were plated on Cell-Tak (Corning, Cat# CB-40240) coated 25 mm coverslips then dye-loaded in a serum-free extracellular medium (ECM, 121 mM NaCl, 5 mM NaHCO$_3$, 10 mM Na-HEPES, 4.7 mM KCl, 1.2 mM KH$_2$PO4, 1.2 mM MgSO$_4$, 2 mM CaCl$_2$, and 10 mM glucose, pH 7.4) containing 2% BSA (Roche). For measurements of $[Ca^{2+}]_{cyto}$, cells were loaded with 2 µM fura2/AM (Invitrogen) or 2 µM fluo8/AM (TEFLabs) in the presence of 0.003% Pluronic F-127 and 100 µM sulfinpyrazone for 10 min at 37 °C. Caged IP$_3$ (D-23-O-Isopropylidene-6-O-(2-nitro-4,5-dimethoxy)ben-zyl-myo-Inositol 145-trisphosphate-Hexakis(propionoxymethyl) Ester, (SiChem, cag-iso-2-145-100) was loaded under the same conditions for 30 min Subsequently, the cells were washed with fresh ECM containing 0.25% BSA and transferred to the thermostated stage (37 °C) of the microscope.

Two imaging systems were used for Ca²⁺ measurements: (1) a ProEM1024 EMCCD camera (Princeton Instruments) fitted to a Leica DMI6000B inverted microscope with Lambda DG4 wavelength-switch xenon light source (Sutter Instruments) and operated by a custom software (Spectralyzer), and (2) an ImagEM EM-CCD camera (Hamamatsu) fitted to an Olympus IX81 microscope with Lambda DG4 light source and Mosaic targeted illumination module (Andor) operated by the Metamorph software (Molecular Devices). Uncaging light was provided by a mercury arc lamp (longpass UV filter, 400 nm cutoff) or a 365 nm LED source (Sutter Lambda TLED+). Both imaging setups used the UV-optimized Olympus UAPO/340 ×40/1.35N.A. oil immersion objective.

For simultaneous measurements of $[Ca^{2+}]_{cyto}$ and $[Ca^{2+}]_{mito}$ was measured fura2 fluorescence was recorded at 340 and 380 nm excitation while fluo8 at 485/15 nm excitation and the mitochondrial matrix-targeted RCaMP was either excited at 545/20 nm (w/fura2) or at 577/20 nm (w/fluo8), using dual-band dichroic and emission filters (Chroma, set 73100 or customized set 59022, respectively). Images were collected every 0.2 s or 2 s. $[Ca^{2+}]_{cyto}$ measurements with fura2 were calibrated in vitro with adding 1 mM CaCl$_2$, followed by 10 mM EGTA/Tris, pH 8.5. To determine $[Ca^{2+}]$, the following formula was used: $[Ca^{2+}] = K_d \times (S_{f2}/S_{b2}) \times (R - R_{min})/(R_{max} - R)$, where K$_d$ is the Ca²⁺-dissociation constant (0.224 µM), R is the ratio of the fluorescence intensities at 340/380 nm excitation, R$_{min}$ and R$_{max}$ are the fluorescence ratios in Ca²⁺-free and Ca²⁺-saturated conditions respectively, S$_{f2}$/S$_{b2}$ is the ratio of fluorescence intensities measured at 380 nm excitation in Ca²⁺-free (f)/Ca²⁺-saturated (Ca²⁺-bound, b) conditions[49].

$[Ca^{2+}]_{cyto}$ and $[Ca^{2+}]_{mito}$ were recorded in permeabilized cells with Fura2FA added to the bulk buffer after permeabilization and rhod2AM accumulated by the mitochondria. One micromolar CaCl$_2$ was added to optimize ER Ca²⁺ loading. IP$_3$ receptors were stimulated with adding IP$_3$ in a final concentration of 7.5 µM.

**FRET measurements**. DT40 cells were co-transfected with ER membrane (CFP-FRB-9×-ER) and OMM (AKAP1-18×-FKBP-YFP) targeted rapamycin-inducible linker pairs. After 24 h cells were plated on Cell-Tak coated coverslips. CFP (435 excitation/480 nm emmision), YFP (500/535 nm) and FRET (435/535 nm) fluorescence signals were measured in ECM containing 0.25% BSA with Leica DMI 6000B inverted microscope at 37 °C. Covalent linkage fromation between the FKBP and FRB domains was induced by adding 100 nM rapamycin.

**Measurements of $[Ca^{2+}]_{cyto}$ in permeabilized cells**. DT40 cells in suspension were pre-incubated in Ca²⁺-free extracellular buffer (120 mM NaCl, 20 mM Na–HEPES, 5 mM KCl, 1 mM KH$_2$PO4, 100 µM EGTA/Tris pH 7.4) for 1 h at 37 °C to drain Ca²⁺ from intracellular compartments and stored on ice. Cells were permeabilized with saponin (40 µg/ml) and incubated in an intracellular medium (ICM; 120 mm KCl, 10 mm NaCl, 1 mm KH$_2$PO4, 20 mm Tris-HEPES, 2 mm Mg-ATP, and antipain, leupeptin, and pepstatin 1 µg/ml each at pH 7.2) and to measure $[Ca^{2+}]_{cyto}$, 1.5 µM fura2/FA was added. Fluorescence was monitored in a fluorometer (Delta-RAM, PTI) using 340 nm and 380 nm excitation and 500 nm emission for fura2. Calibration of the fura2 was carried out at the end of each measurement adding 1 mM CaCl$_2$, followed by 10 mM EGTA/Tris, pH 8.5 and using the formula described above[49].

**Immunocytochemistry**. DT40 Cells were transfected with ER-lumen targeted GFP and OMM-targeted mCherry (OMP25-mCherry) expression vectors. Transfected cells were incubated at 40 °C for 24 h then attached to Cell-Tak treated coverslips and further incubated for 4 h. Cells were fixed in 3% paraformaldehyde/PBS and labelled with anti-FLAG (1:500 dilution, Sigma–Aldrich Cat# F1804) or anti-IP3R1 primary antibody (CT1)[36] then AlexaFluor 647-conjugatedanti-mouse or anti-rabbit secondary antibody (1:200 dilution, Thermo Fisher, Cat# A-21239 and

A-21244). Immunofluorescence was imaged using a Zeiss LSM 880 confocal system equipped with Airyscan detector, using 488, 561, and 633-nm laser lines for green, red and far-red fluorescent probes, respectively. Images were taken through a ×63 oil objective (numerical aperture [NA] 1.40). Co-distribution of ER and mitochondria was calculated as the log2 ratio of the cumulative abundance of mitochondria and that of the ER in the 0–1 µm Z sections of each cells.

**Superresolution localization microscopy**. 3D super-resolution microcopy was performed using a Vutara 352 nanoscope (Bruker). Briefly, two focal planes were projected onto the sensor of an ORCA Flash v.2 sCMOS camera (Hamamatsu) through an Olympus UPlanSApo ×60/1.2 N.A. water-immersion objective. The point spread functions for the wavelengths used were calibrated and aligned using 100 nm Tetraspeck microspheres (ThermoFisher Scientific). Laser illumination at 640 and 561 nm were used for stochastic imaging of AlexaFluor 647 and CF-568 (Biotium), respectively, and 405 nm light was used to recover portions on CF568 from their dark state. The two colors were recorded sequentially (AF647, then CF568) and several thousand frames were acquired for each fluorophore. Cells were plated, fixed and IP3Rs were labelled as described above, and TOM20 was labelled with anti TOM20 (dilution: 1:200, Santa Cruz Cat# sc-11415) primary then with CF568 Goat Anti-Rabbit IgG (dilution: 1:200 Biotium Cat# 20103-1) secondary antibodies. Imaging was performed in freshly prepared buffer for oxygen-depleting and reducing conditions: 20 mM cysteamine, 1% 2-mercaptoethanol (v/v), 169 AU/ml glucose oxidase and 1400 AU/ml catalase in 50 mM Tris-HCl, 10 mM NaCl and 10% (w/v) glucose (pH 8.0). Calibration, alignment, acquisition and localization were all done in the Vutara SRX software.

Cluster-based minimum distance analysis was also performed in Vutara SRX. Empirically determined settings were used to generate IP3R clusters (minimum particles: 6, particle size: 70 nm, accumulation threshold: 0.04) and large clusters that represented the volumes of whole mitochondria (minimum particles: 100, particle size: 250 nm, accumulation threshold: 0.05), which were used for all measurements. Control data consisted of randomly generated clusters containing 20 particles each (30 and 75 nm FWHM lateral and axial, respectively) at a density of 12.5 clusters/µm³.

**3D STED microscopy**. 3D STED microscopy was performed using an Abberior Instruments Expert Line STED microscope equipped with an Olympus UPLSAPO ×100/1.4NA oil immersion objective. Briefly, Tom20 was labelled with anti-TOM20 primary (dilution: 1:200, Proteintech, Cat#11802-1-AP) and Alexa Fluor 594 anti-rabbit IgG secondary (dilution: 1:150, Molecular Probes Cat#A-11037) antibodies. Flag-tagged IP3Rs were labelled with anti-FLAG primary (dilution: 1:500, Sigma–Aldrich Cat# F1804) and STAR RED, goat anti-mouse IgG (dilution: 1:150, Abberior, Cat#STRED-1001-500UG) secondary antibodies. Sequential confocal and STED images were obtained following excitation of Alexa Fluor 594 and STAR RED by 594 and 640 nm lasers, respectively. Both fluorophores were depleted in three dimensions with a 775 nm pulsed STED laser. Z-stacks were obtained by collecting images at 150 nm intervals using the 3D STED mode. RESCue STED was employed to minimize the light dosage. In this mode when the photon count returned in the confocal images was below 10, the STED laser was not engaged for that pixel. Blend mode depth projection images were generated and fluorophore volumes and interfaces between these volumes were analyzed using IMARIS 9.2.1 (Bitplane).

**Statistical analysis**. For the statistical tests Microsoft Excel or Systat SigmaPlot softwares were used. Significance was tested with two-tailed tests. Normality was tested with Systat SigmaPlot. Datasets of two independent groups were pairwise compared with unpaired Student's $t$-test or Mann–Whitney Rank Sum Test if normality test failed. Multiple samples were compared with one-way ANOVA with post-hoc Holm method (all vs. control) or post-hoc Dunn's method (all pairwise). Difference between the expected frequencies and the observed were tested with Chi-squared test.

All graphs were created in SigmaPlot 12 and were combined with the images in PowerPoint.

Every element of each graph and image was created by the Authors. No previously created elements were used.

**Reporting summary**. Further information on research design is available in the Nature Research Reporting Summary linked to this article.

## Data availability
The datasets generated during and/or analyzed during the current study are available from the corresponding author on reasonable request.

## Code availability
Custom image analysis software (Spectralyzer) and macros prepared for ImageJ are available from the corresponding author on reasonable request. The ImageJ plugin for analysis of ER-mitochondrial interfaces in TEM images is available from the update site: http://sites.imagej.net/MitoCare/

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

## Acknowledgements

We thank Dr. T. Kurosaki for providing us with DT40 cells. Adam Bartok was supported by a Hungarian State Eotvos Fellowship from the Tempus Public Foundation (Hungary). This study was funded by NIH grants DK051526 and GM059419 (G.H.), ES025672 (G.H. and G.C.), DK103558 (S.K.J. and G.H.), and DE014756 (DIY). The Abberior STED microscope was partially funded by NIH S10ODO23440 (D.I.Y.).

## Author contributions

Conceptualization: G.H., G.C., S.K.J., D.I.Y. and A.B.; Investigation: A.B., D.W., T.G., M.K., Z.N., S.B., V.K.T., K.A., H.A., K.M. and G.C.; Writing: G.H., A.B., D.W., G.C., S.K.J., D.I.Y. and; Funding Acquisition: G.H., G.C., S.K.J., D.I.Y. and A.B.

## Additional information

**Competing interests:** The authors declare no competing interests.

