## [Peer Review File · Nature Communications]

Reviewers' comments:

Reviewer #1 (Remarks to the Author):

The paper by Bartok et al. proposes a crucial role of IP3Rs in maintaining the ER-mitochondria physical association. Using WT and IP3Rs triple KO chicken DT40 B cell lines, the authors described a novel function of IP3Rs, which is independent of Ca²⁺ fluxes and confers resistance against oxidant-induced cell stress. They conclude that IP3Rs possess an intrinsic structural activity, shading light also on the role of type 2 IP3R in the ER-mitochondria Ca²⁺ transfer. Although this study is novel and potentially interesting, I have a number of important issues that undermine confidence in the conclusions.

The major limitation is that all the experiments have been performed only in DT40 cells. Due to the relevance of the message, the structural role of IP3Rs has to be confirmed in at least other two cell lines. Moreover, I have some concerns about data quality and consistency. For example, in some experiments the authors used two IP3R1-expressing clones (R1A and B), whereas in other figures only a single clone has been analysed. In figure 2, the airyscan plots of TKO cells is missing and I am wondering whether a DT40 TKO control stable clone has been created to compare the data obtained in R1-, R2- and R3-expressing DT40 stable clones. Again, a western blot showing the levels of different IP3Rs is missing.

In figure 6 the abilities of inactive (pore dead) IP3R2 and 3 mutants in re-establishing normal ER-mitochondria contacts have not been investigated. This represents a very important aspect, since the authors affirm that different IP3R isoforms induced Ca²⁺ signal from the ER to the mitochondria with a very different efficacy.

Finally, the association between the structural role of IP3Rs and their capacity in restoring cell viability (Fig. 7 and schematic model in Fig. 8) requires additional clarifications. Does oxidant stress induce ER-mitochondria contacts remodelling?

Overall, this is a potentially interesting story, but with significant weakness as well, which requires additional controls and experimental evidence.

Reviewer #2 (Remarks to the Author):

The manuscript by Bartok, et.al., investigates the function of IP3 receptor in ER-mitochondrial contacts and local calcium transfer. The authors have reached the following major conclusions: IP3Rs is necessary to form a tight contact between ER and mitochondria and each IP3R isoform can tighten

the interactions between the ER and mitochondria, but their intracellular distributions have distinct difference. The authors used single molecule localization microscopy to confirm the close contacts between IP3R isoforms and mitochondria and to reveal distinct difference of intracellular distribution of each IP3R isoform. Through confocal with Airy scan and TEM imaging, the authors also discovered that TKO cells with dead pore of IP3R1 can also support the ER-mitochondrial contact as well as resist from H₂O₂ induced cell death, indicating that IP3 receptors have structural and protective functions that are independent of Ca²⁺ flux through the receptors.

The manuscript is well written and descriptions on experiments performed are clear. However, the experiments and quantifications of protein abundance and co-localization based confocal and super-resolution data do not have enough resolution to support several of the major claims in the manuscript. At this stage, it is unclear whether the authors will be able to support their conclusions with further evidence. Therefore, we cannot recommend publication of the manuscript at this stage. The major concerns are listed below:

Major comments:

1. One main technique the authors used to visualize and quantify the ER-mitochondria contact sites are from confocal imaging with an Airy scan detector. However, the resolution achieved is not enough to confirm cellular positioning of tagged protein or its co-localization with the mitochondria membrane marker. Specifically, the method of calculating the relative abundance at a set of z positions suggests that the fluorescence signals from ER and mitochondria have similar relative total intensity at each z position which cannot be used as a quantitative evidence of actual contact. And two-color overlay images suggest that ER and mitochondria locate at a similar location within the z slice. It failed to serve as an evidence that there are actual contacts.

2. The results from single molecule localization of IP3R isoforms and OMM did not achieve enough resolution in resolving TOM20 clusters and IP3Rs and therefore making the following quantification less convincing. We suggest to further optimize experimental conditions of two-color super-resolution imaging which, in principle, should provide enough 3D resolution in resolving distinctly the tagged clusters.

3. In our opinion, TEM and FRET experiments in the manuscript provide supporting evidence (however, not direct evidence) to the conclusions listed in the manuscript. No results based on TEM and FRET measurements related to the interactions between IP3R isoforms and mitochondria are shown.

Minor comments:

1. Is "Fig 7C" in line 265 referring to "Fig 7B"?
2. We cannot find legend in Fig2. BCD.
3. In Fig 7B, there seems to be no difference between TKO cells and the other conditions. It is unclear to the reviewer what is the conclusion to be drawn from this figure.

Reviewer #3 (Remarks to the Author):

This is a comprehensive study of the role of the different isoforms of the IP3R as structural components of the ER-mitochondria interface. By using careful systems of knockout and reintroduction, the authors clarify that each isoform can increase the proximity between the two organelles, measured with an array of approaches (from superresolution to FRET sensors to EM). Functionally, the authors identify that the role of the IP3R in tethering is Ca²⁺ independent and that all of the different isoforms can transfer Ca²⁺ to mitochondria, R2 being the most efficient one.

In general, this manuscript is of great interest for the booming field of contact sites between organelles. The study is performed at a very high technical level and the conclusions are very well supported by the data shown. I only found the part on cell death marginal to the central question of the paper and suggest that the authors remove it from the final version of the manuscript to be published.

Reviewers' comments:

Reviewer #1 (Remarks to the Author):

The paper by Bartok et al. proposes a crucial role of IP3Rs in maintaining the ER-mitochondria physical association. Using WT and IP3Rs triple KO chicken DT40 B cell lines, the authors described a novel function of IP3Rs, which is independent of Ca²⁺ fluxes and confers resistance against oxidant-induced cell stress. They conclude that IP3Rs possess an intrinsic structural activity, shading light also on the role of type 2 IP3R in the ER-mitochondria Ca²⁺ transfer. Although this study is novel and potentially interesting, I have a number of important issues that undermine confidence in the conclusions.

The major limitation is that all the experiments have been performed only in DT40 cells. Due to the relevance of the message, the structural role of IP3Rs has to be confirmed in at least other two cell lines.

The Mikoshiba group has recently created HeLa cells lacking each IP3 receptor isoform (triple knockout, TKO) (PMID:30429331) and we performed quantitative ultrastructural analysis of the ER-mitochondrial contacts in both wild type and TKO HeLa cells. The new results are included in the revised manuscript as Fig1F and show less tight contacts in the TKO than wild type HeLa consistent with what we have shown for the TKO and wild type DT40 cells (Fig1E). Because adding one new cell line required a major effort, and the Editor stated that one additional cell line is acceptable for the journal, we respectfully ask the Reviewer to accept this extension of the original model satisfactory.

Moreover, I have some concerns about data quality and consistency. For example, in some experiments the authors used two IP3R1-expressing clones (R1A and B), whereas in other figures only a single clone has been analysed.

Our idea behind using two distinct clones of IP3R1 was to confirm for some of the parameters that no clonal variability is behind the differences observed between the TKO and rescue clones. Using two IP3R1 clones we can state that the differences described in the IP3 sensitivities, intracellular localization and interorganellar signal transmission do not arise from the possible differences in the protein expression levels (Fig S1A) in the stable clones but specific phenotypic aspects of the different isoforms expressed in functionally similar levels (see IP3 sensitive ER pool sizes). Based on this observation we omitted using both clones in experiments that require expensive materials (e.g. caged IP3 complex).

Notably, in the revised ms we also included measurements using DT40 clones in which 2 out of the 3 IP3R isoforms were deleted (Fig S3). These double-KO cells allowed us to study each endogenous isoform in isolation. The results confirmed that the endogenous avian IP3R2 is highly efficient in ER-mitochondrial Ca²⁺ transfer in DT40 cells like the mammalian IP3R2 used for the rescue (pg8 last para, pg9 first para).

In figure 2, the airyscan plots of TKO cells is missing and I am wondering whether a DT40 TKO control stable clone has been created to compare the data obtained in R1-, R2- and R3-expressing DT40 stable clones.

Data for WT, TKO and stable rescued cells were obtained side by side in several independent experiments. We considered that the description of our findings can be most effectively presented by introducing the most striking structural findings in a WT vs TKO comparison, followed by the description of the stable rescue cells. Thus, distribution of ER and mitochondria in the z-stacks for TKO is shown only in Fig1 to avoid redundancy. However, we display the TKO in every bar charts that show the quantitative analysis of Airyscan and electron microscopy (see Figs 2 and 6).

In addition, we created for the Reviewer and show below a version of the Airyscan results of Figs2&6, which includes the TKO, too.

As a “control stable TKO clone” we created TKO DT40 cells with expression of the M3 muscarinic receptor. As shown in Fig6GH, this cell line behaves as TKO cells both in regards of ER-mitochondrial associations and IP3-linked Ca²⁺ mobilization

Again, a western blot showing the levels of different IP3Rs is missing.

In the original submission, instead of measuring protein expression levels with WB, we compared the different isoform expressing clones with functional measurements (Fig S1) where the functional level of the IP3Rs was measured with the Ca²⁺ mobilization ability upon saturating IP3R stimulus (pool size). To complement this analysis with direct assessment of the protein levels, we performed and included the requested western blot experiments (Fig S1A,B).

In figure 6 the abilities of inactive (pore dead) IP3R2 and 3 mutants in re-establishing normal ER-mitochondria contacts have not been investigated. This represents a very important aspect, since the authors affirm that different IP3R isoforms induced Ca²⁺ signal from the ER to the mitochondria with a very different efficacy.

IP3R1 D2550A was the only stable clone in DT40 cells that carries a non-functional pore. However, recently we produced a stable clone of R2 GS3 DT40 clone, validated that this harbors an inactive channel mutant (Fig 6G). We have also confirmed that this inactive R2 has the ability to restore the tight connections between the ER and mitochondria (Fig6H). As of now we have no knowledge of described and characterized pore mutant R3 receptors.

Finally, the association between the structural role of IP3Rs and their capacity in restoring cell viability (Fig. 7 and schematic model in Fig. 8) requires additional clarifications.

Does oxidant stress induce ER-mitochondria contacts remodelling?

Because systematic investigation of the effect of oxidant stress would require a great deal of additional work diverting the focus from physiology to pathophysiology and because Reviewer #3 suggested to omit the component on oxidants, we eliminated this from the revised ms.

We revised the scheme to accurately reflect our findings.

Overall, this is a potentially interesting story, but with significant weakness as well, which requires additional controls and experimental evidence.

We appreciate the Reviewer's thoughtful and encouraging comments and hope that our additional studies and revising of the manuscript address the weaknesses.

Reviewer #2 (Remarks to the Author):

The manuscript by Bartok, et.al., investigates the function of IP3 receptor in ER-mitochondrial contacts and local calcium transfer. The authors have reached the following major conclusions: IP3Rs is necessary to form a tight contact between ER and mitochondria and each IP3R isoform can tighten the interactions between the ER and mitochondria, but their intracellular distributions have distinct difference. The authors used single molecule localization microscopy to confirm the close contacts between IP3R isoforms and mitochondria and to reveal distinct difference of intracellular distribution of each IP3R isoform. Through confocal with Airy scan and TEM imaging, the authors also discovered that TKO cells with dead pore of IP3R1 can also support the ER-mitochondrial contact as well as resist from H₂O₂ induced cell death, indicating that IP3 receptors have structural and protective functions that are independent of Ca²⁺ flux through the receptors.

The manuscript is well written and descriptions on experiments performed are clear. However, the experiments and quantifications of protein abundance and co-localization based confocal and super-resolution data do not have enough resolution to support several of the major claims in the manuscript. At this stage, it is unclear whether the authors will be able to support their conclusions with further evidence. Therefore, we cannot recommend publication of the manuscript at this stage. The major concerns are listed below:

Major comments:

1. One main technique the authors used to visualize and quantify the ER-mitochondria contact

sites are from confocal imaging with an Airy scan detector. However, the resolution achieved is not enough to confirm cellular positioning of tagged protein or its co-localization with the mitochondria membrane marker. Specifically, the method of calculating the relative abundance at a set of z positions suggests that the fluorescence signals from ER and mitochondria have similar relative total intensity at each z position which cannot be used as a quantitative evidence of actual contact. And two-color overlay images suggest that ER and mitochondria locate at a similar location within the z slice. It failed to serve as an evidence that there are actual contacts.

Confocal imaging with Airyscan detector was used to visualize intracellular localization of ER, mitochondria and IP3Rs. Airyscan detector increases the optical resolving power to approx 150 nm. In our experiments due to the small size of the DT40 cells (approx 10 μ M diameter) and their relative large, central nucleus we observed that the organelles are tightly arranged as the result of the spatial limitations. Therefore, we could not analyze colocalization of the ER and mitochondria. Instead, we found the observed differences in the relative Z distribution of the organelles rather interesting and worth mentioning, considering that these cells live normally in suspension, the attachment to a surface creates anisotropy along the z axis. Understanding that the optical resolution does not support quantification of colocalization, we chose to use the word co-distribution to describe the nature of our findings. To obtain “quantitative evidence of actual contacts” the higher resolution techniques were applied, such as FRET and TEM. In the revision, we expanded TEM, and included STED and immune-TEM (Fig S2, pg7 para3,4, pg8 para1).

2. The results from single molecule localization of IP3R isoforms and OMM did not achieve enough resolution in resolving TOM20 clusters and IP3Rs and therefore making the following quantification less convincing. We suggest to further optimize experimental conditions of two-color super-resolution imaging which, in principle, should provide enough 3D resolution in resolving distinctly the tagged clusters.

We should have been clearer in our description of the cluster-based measurements from single molecule localization imaging. As applied to the TOM20 staining, we were not seeking to define clusters of the protein, per se, as was done by Wurm et al. (PNAS 2011), but rather we tried to generate the surface of the entire organelles as large clusters (containing >500 individual localization events, compared with the IP3R clusters with a minimum of 6 localizations). This approach also served as a kind of filtering mechanism to eliminate detections that were presumed to arise from non-specific antibody binding. Further, we purposely adjusted our imaging and analysis conditions to achieve a filled-in appearance of the OMM, however if we eliminate lower confidence localizations to thin the rendered image, we are able to see a more clustered appearance of the TOM20. The corresponding section of the Results has been rewritten for greater clarity and a paragraph has been added to the Methods giving detailed explanation of the

minimum distance analysis (pg17 para2).

Regarding the resolution achieved in these imaging experiments, using the method described by Niewenhuizen et al. (Nat. Methods 2013) as implemented in the Vutara SRX software, the typical resolution of our images is estimated at 50-60 nm in the x-y dimensions and 70-90 nm in z.

3. In our opinion, TEM and FRET experiments in the manuscript provide supporting evidence

(however, not direct evidence) to the conclusions listed in the manuscript. No results based on TEM and FRET measurements related to the interactions between IP3R isoforms and mitochondria are shown.

Fig2D shows TEM evaluation of DT40 cells expressing each IP3R isoform and provides evidence that each IP3R isoform is competent to enhance (restore in TKO) tight associations between ER and mitochondria. Furthermore, the immunoTEM results of the revised ms (FigS2) provide evidence for enrichment of IP3Rs, in particular IP3R2 in the <25nm vicinity of the mitochondria.

Minor comments:

1. Is "Fig 7C" in line 265 referring to "Fig 7B"? Yes, corrected

2. We cannot find legend in Fig2. BCD. The legend is on pg21.

3. In Fig 7B, there seems to be no difference between TKO cells and the other conditions. It is unclear to the reviewer what is the conclusion to be drawn from this figure. In response to the request of Reviewer #3 we removed the results on oxidant stress.

We appreciate the thoughtful suggestions and the positive comments of the Reviewer.

Reviewer #3 (Remarks to the Author):

This is a comprehensive study of the role of the different isoforms of the IP3R as structural components of the ER-mitochondria interface. By using careful systems of knockout and reintroduction, the authors clarify that each isoform can increase the proximity between the two organelles, measured with an array of approaches (from superresolution to FRET sensors to EM). Functionally, the authors identify that the role of the IP3R in tethering is Ca²⁺ independent and that all of the different isoforms can transfer Ca²⁺ to mitochondria, R2 being the most efficient one.

In general, this manuscript is of great interest for the booming field of contact sites between organelles. The study is performed at a very high technical level and the conclusions are very well supported by the data shown. I only found the part on cell death marginal to the central question of the paper and suggest that the authors remove it from the final version of the manuscript to be published.

We thank the Reviewer for her/his suggestion and for the positive comments. We have removed the cell death results from the revised ms as suggested.

Reviewers' comments:

Reviewer #1 (Remarks to the Author):

The revised version of this manuscript is significantly improved. The authors performed additional experiments that effectively solved my original criticisms. I think the manuscript now deserves publication.

Reviewer #2 (Remarks to the Author):

Bartok et al., revised the manuscript with extensive clarifications and additional dataset which helped to clarify our previous concerns.

However, I remain unsure about several conclusions drawn from z-distribution comparisons based on confocal data. For example, authors conclude, as in section starting with line 165, "These results support the hypothesis that each IP3R isoform enhances the physical connection between the ER and mitochondria."

Z projection distribution changes of ER and mitochondria marker distribution (from line 165-line 176) provide evidence for IP3R's effect on redistribution of cellular organelles but the redistribution can not provide evidence towards the above conclusion on "physical connection". It would be helpful if the authors could clarify this further.

In general, it is uncertain to me that how much conclusion one could draw from these z-distribution data presented throughout this work. This concern is related to my original comment #1.

Reviewers' comments:

Reviewer #1 (Remarks to the Author):

The revised version of this manuscript is significantly improved. The authors performed additional experiments that effectively solved my original criticisms. I think the manuscript now deserves publication.

We appreciate the Reviewer's support of the publication of our ms.

Reviewer #2 (Remarks to the Author):

Bartok et al., revised the manuscript with extensive clarifications and additional dataset which helped to clarify our previous concerns.

However, I remain unsure about several conclusions drawn from z-distribution comparisons based on confocal data. For example, authors conclude, as in section starting with line 165, "These results support the hypothesis that each IP3R isoform enhances the physical connection between the ER and mitochondria."

Z projection distribution changes of ER and mitochondria marker distribution (from line 165-line 176) provide evidence for IP3R's effect on redistribution of cellular organelles but the redistribution can not provide evidence towards the above conclusion on "physical connection". It would be helpful if the authors could clarify this further.

In general, it is uncertain to me that how much conclusion one could draw from these z-distribution data presented throughout this work. This concern is related to my original comment #1.

We appreciate the Reviewer's thoughtful suggestion about clarifying the conclusions we draw from the different assays of organellar morphology.

We have carefully checked the entire manuscript and rephrased and further clarified every statement that might have been understood as an overinterpretation of the z-distribution data in terms of ER-mitochondrial associations.

See lines 140-141, 166-172, 297-299, 326-345, 354-356, 588-589.

REVIEWERS' COMMENTS:

Reviewer #2 (Remarks to the Author):

In this revision, the authors addressed my previous concern. I believe the manuscript is now suitable for publication.